# Gate-based quantum computing for protein design

**Mohammad Hassan Khatami**[1], Udson C. Mendes[2], Nathan Wiebe[3,4,5], Philip M. Kim[1,3,6]*

**1** Terrence Donnelly Centre for Cellular & Biomolecular Research, University of Toronto, Toronto, Ontario, Canada, **2** CMC Microsystems, Sherbrooke, Québec, Canada, **3** Department of Computer Science, University of Toronto, Toronto, Ontario, Canada, **4** Pacific Northwest National Laboratory, Richland, Washington, United States of America, **5** Department of Physics University of Washington, Seattle, Washington, United States of America, **6** Department of Molecular Genetics, University of Toronto, Toronto, Ontario, Canada

* pi@kimlab.org

**Data Availability Statement:** All relevant data are within the paper and its Supporting Information file. The codes to create the circuits in the MR and SP models in this study are available on Zenodo at https://zenodo.org/record/7344649 and are

## Abstract

Protein design is a technique to engineer proteins by permuting amino acids in the sequence to obtain novel functionalities. However, exploring all possible combinations of amino acids is generally impossible due to the exponential growth of possibilities with the number of designable sites. The present work introduces circuits implementing a pure quantum approach, Grover's algorithm, to solve protein design problems. Our algorithms can adjust to implement any custom pair-wise energy tables and protein structure models. Moreover, the algorithm's oracle is designed to consist of only adder functions. Quantum computer simulators validate the practicality of our circuits, containing up to 234 qubits. However, a smaller circuit is implemented on real quantum devices. Our results show that using $\mathcal{O}(\sqrt{N})$ iterations, the circuits find the correct results among all $N$ possibilities, providing the expected quadratic speed up of Grover's algorithm over classical methods (i.e., $\mathcal{O}(N)$).

## Author summary

Protein design aims to create novel proteins or enhance the functionality of existing proteins by tweaking their sequences through permuting amino acids. The number of possible configurations, $N$, grows exponentially as a function of the number of designable sites ($s$), i.e., $N = A^s$, where $A$ is the number of different amino acids ($A = 20$ for canonical amino acids). The classical computation methods require $\mathcal{O}(N)$) queries to search and find the low-energy configurations among $N$ possible sequences. Searching among these possibilities becomes unattainable for large proteins, forcing the classical approaches to use sampling methods. Alternatively, quantum computing can promise quadratic speed-up in searching for answers in an unorganized list by employing Grover's algorithm. Our work shows the implementation of this algorithm at the circuit level to solve protein design problems. We first focus on lattice model-like systems and then improve them to more realistic models (change in the energy as a function of distances). Our algorithms can implement various custom pair-wise energy tables and any protein structure models.

available on GitHub at https://github.com/ Mohammad-Khatami/grover-protein-desing.

**Funding:** PMK received a Canadian Institutes of Health Research (CIHR) grant PJT-159750, which supported this project. (https://cihr-irsc.gc.ca) MHK's salary was partially funded by Canadian Institutes of Health Research (CIHR) grant PJT-159750. The funders had no role in study design, data collection and analysis, decision to publish, or preparation of the manuscript.

**Competing interests:** I have read the journal's policy and the authors of this manuscript have the following competing interests: UCM is the leader of the quantum computing team at CMC Microsystems. PMK is a co-founder and consultant to multiple companies, including Oracle Therapeutics and TBG Therapeutics, and serves on the scientific advisory board of ProteinQure. MHK and NW declare no Competing Financial or Non-Financial Interests.

We have used quantum computer simulators to validate the practicality of our circuits which require up to 234 qubits. We have also implemented a simple version of our circuits on real quantum devices. Our results show that our circuits provide the expected quadratic speed-up of Grover's algorithm.

This is a *PLOS Computational Biology* Methods paper.

## 1. Introduction

Protein design is a procedure to construct proteins with certain configurations to achieve novel functionality. In this regard, amino acids are mutated in the protein's sequence to find sets of residues that provide the lowest energy of the protein in the expected configuration. Using computational approaches, one could consider having "$s$" designable sites in the sequence and "$A$" different amino acids that could fill these sites, where $A = 20$ for the canonical amino acids. This will provide $A^s$ possible sets of amino acids to find the answer. Thus, the number of possible sets of amino acid sequences grows exponentially by increasing the designable sites.

In computer science terminology, the protein design is categorized under non-deterministic polynomial-time (NP)-hard problems [1,2]. The main characteristics of these problems are that the computational time and resources needed to find their solutions scale poorly. Thus, finding their answer using conventional computers could become either impossible or requires a great deal of computational time. However, the validity of a proposed answer could be evaluated in polynomial time by conventional computers [3]. Even for the simple hydrophobic-polar (HP) [4] protein lattice models containing only two types of residues (hydrophobic and polar), the protein design is shown to be in the class of NP-hard problems [5].

Statistical methods such as Markov chain Monte Carlo (MCMC) [6,7] are currently being used to solve NP-hard problems, including protein design problems, on conventional computers [8–10]. In these methods, the algorithm uses sampling techniques and probability distributions to find the answers among all possible sets of amino acids. However, since the probabilistic methods do not explore all sets, it is possible to miss some of the answer states.

Unlike conventional approaches, quantum computation techniques are expected to enhance solving the NP class of problems in their exact forms [3]. In recent years, there have been attempts to use quantum computers to solve NP-hard problems in protein studies, mainly focused on protein folding [11–14]. In these studies, hybrid quantum-classical algorithms employing gate-based quantum devices, such as the Quantum Approximate Optimization Algorithm (QAOA) [15] and the Variational Quantum Eigensolver (VQE) [16], as well as quantum annealing approaches [17] are implemented.

As an example of gate-based approaches, Fingerhuth *et al.* [14] used the QAOA method to study the protein folding on a 9-residue protein using the *HP* energy model and a 3D lattice structure. Similarly, a version of the VQE approach is employed by Robert *et al.* [13] to study the protein folding of a 10-residue protein, Angiotensin, and a 7-residue neuropeptide. In the case of implementing quantum annealers for protein folding, Perdomo-Ortiz *et al.* [11] studied the folding of a 6-residue peptide using a fixed energy table in a 2D lattice model, employing 81 D-Wave's "superconducting quantum bits". Similarly, Babej *et al.* [12] have worked on folding a ten residue Chignolin protein and an eight residue Trp-Cage peptide in 2D and 3D lattice models, respectively. This study used 2048 superconducting quantum bits of D-Wave's

2000Q quantum annealer device. Regardless of the approach, the studies employing quantum computation for protein folding are mainly limited to peptides with only a handful of residues and a very simplified or limited number of amino acid types, e.g., the *HP* model. In the case of protein design, Mulligan *et al.* [18] have prepared a hybrid quantum-classical solver in Rosetta software [19], called *QPacker*, to address the protein design problem on D-Wave's 2000Q quantum annealer device. Despite the attempts to use quantum computation, to the best of our knowledge, there are no records of studies in which a pure quantum computational method based on a pure gate-based approach is used to investigate protein design problems.

This work introduces a procedure to build gate-based quantum circuits, employing Grover's algorithm [20], to solve protein design problems. Grover's algorithm is a fundamental and famous quantum computation algorithm that offers a quadratic speedup in finding answers in an un-sorted list over classical methods [3,20–22]. In general, Grover's algorithm is composed of four main parts: initialization, Grover's oracle, Grover's diffuser and measurement (Fig 1A). The initialization, the diffuser and the measurement steps are almost similar for all systems in this study. However, the oracle step varies depending on the complexity of the system and is the only step that requires auxiliary work qubits, in addition to the $n$ qubits (Fig 1B).

In the initialization step, $n$ qubits are allocated to create a superposition of $N = 2^n$ quantum states by applying Hadamard (H) gates i.e., $H^{\otimes n}$, representing all possible answer states in the system. This step is similar to other quantum algorithms such as Shor's algorithm [23], and the Deutsch-Jozsa algorithm [24].

In the oracle, the circuit is programmed to implement energies and do all necessary calculations to find the answer states (Fig 1B). This part of the algorithm inputs the general features of the structure (Fig 2), pre-computed pair-wise interaction energy tables (Fig 3), and a threshold energy ($E_{th}$) value (all discussed in detail later in the paper). In addition, pre-computed distance features are provided as input to the oracle depending on the system's complexity. The oracle is designed to use only the summation operation (and multiplication, technically a form of summation). First, the energy values are summed for each pair of interacting designable sites to find the total energy of the sequence. Then, it is subtracted from the $E_{th}$. If the result is negative, the oracle marks that sequence as an answer state by negating its amplitude (Fig 1B). Note that since the probability of each state is the only quantity of a quantum state that could be measured in the classic realm, the negative amplitude of the answer states in the oracle step does not change the relative probabilities of states (*probability* = (*amplitude*). (*amplitude*)*).

In the next step of the algorithm, the diffuser increases the probability of answer states, marked by the oracle, in the circuit (Fig 1A). In the final step, the results of the circuit are measured. To increase the probability of finding the answer states, the oracle and the diffuser steps should be repeated for the $\mathcal{O}(\sqrt{N/M})$ number of iterations (R), where M is the number of answer states in the circuit [3].

Three main oracle models are developed in this study, each with a distinct protein structure representation, distance dependencies, and pair-wise interaction energy tables. In the first model, which is addressed as the "SP" (simplified) model, there are no distance dependencies between the designable sites, and the 2D lattice model-like structures represent the protein (Fig 2A, 2B and 2C). Moreover, only integer numbers are used for energies (Fig 3A), and the oracle only uses the summation function. In the second model, i.e., the "MR" (more realistic) model, the pair-wise energy table (Fig 3B) and pre-computed reciprocals of the distances ($d^{-1}$) between the designable sites are introduced in the circuit (Fig 2D and 2E). First, the distance reciprocals are multiplied by the pair-wise energy of residues filling each designable site. Then, the values are summed to find the system's total energy. In this model, fixed-point decimal

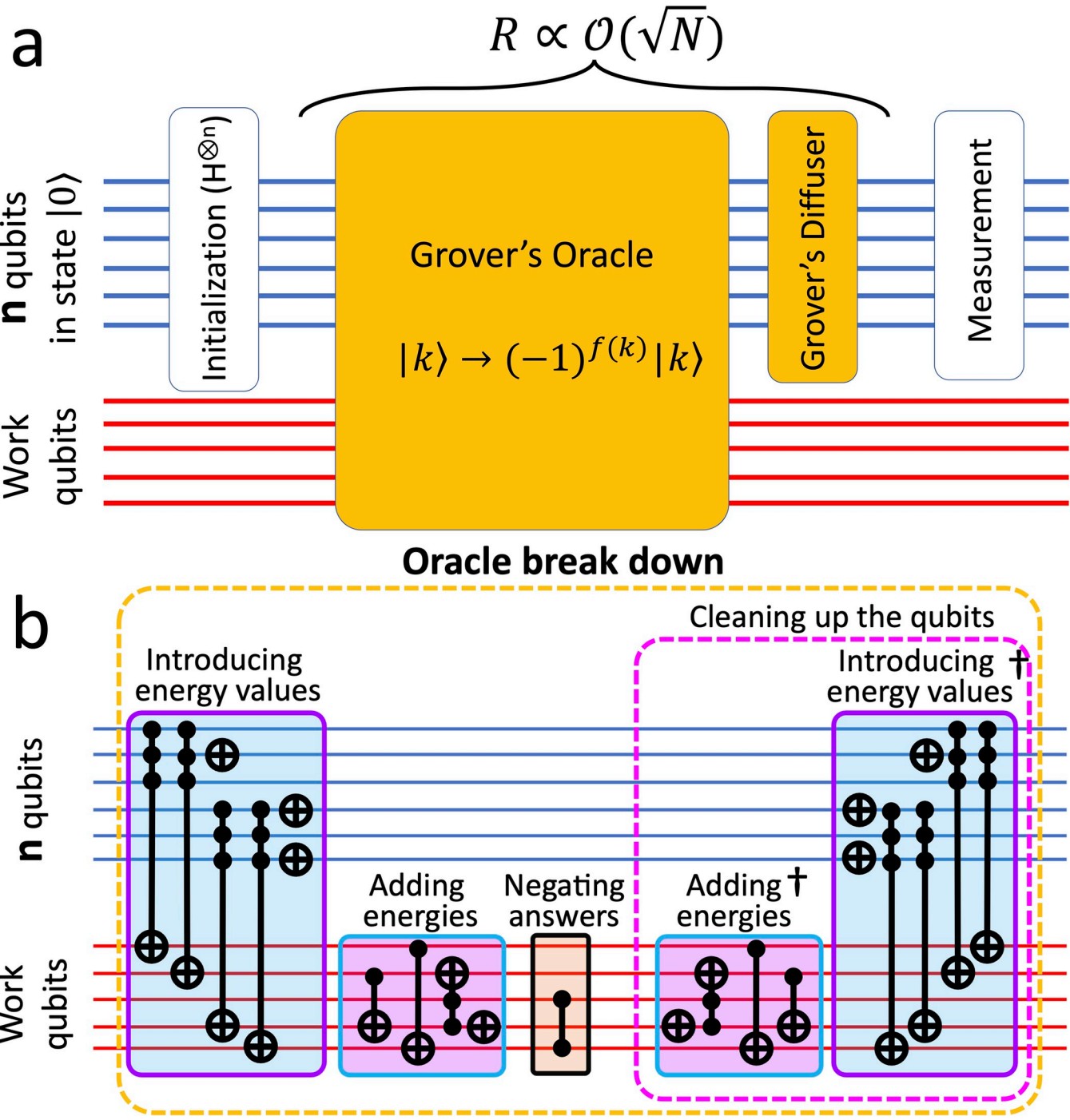

**Fig 1. Schematic representation of our circuits.** A) Different steps of Grover's algorithm; B) Different sub-steps implemented in our Oracle. The parallel lines represent qubits. The blue lines show the $n$ qubits and the red lines show the work qubits. In the oracle, if $|k\rangle$ is an answer state, $f(k) = 1$, otherwise it is 0.

numbers are used to calculate the results (Fig 3B). For the SP and MR models, since we are in the noisy intermediate-scale quantum (NISQ) devices era [25], we use quantum computer simulators to study the circuits' validity and results. However, to test the practicality of our algorithms on real IBM quantum devices, a third model is developed, which is a simplified version

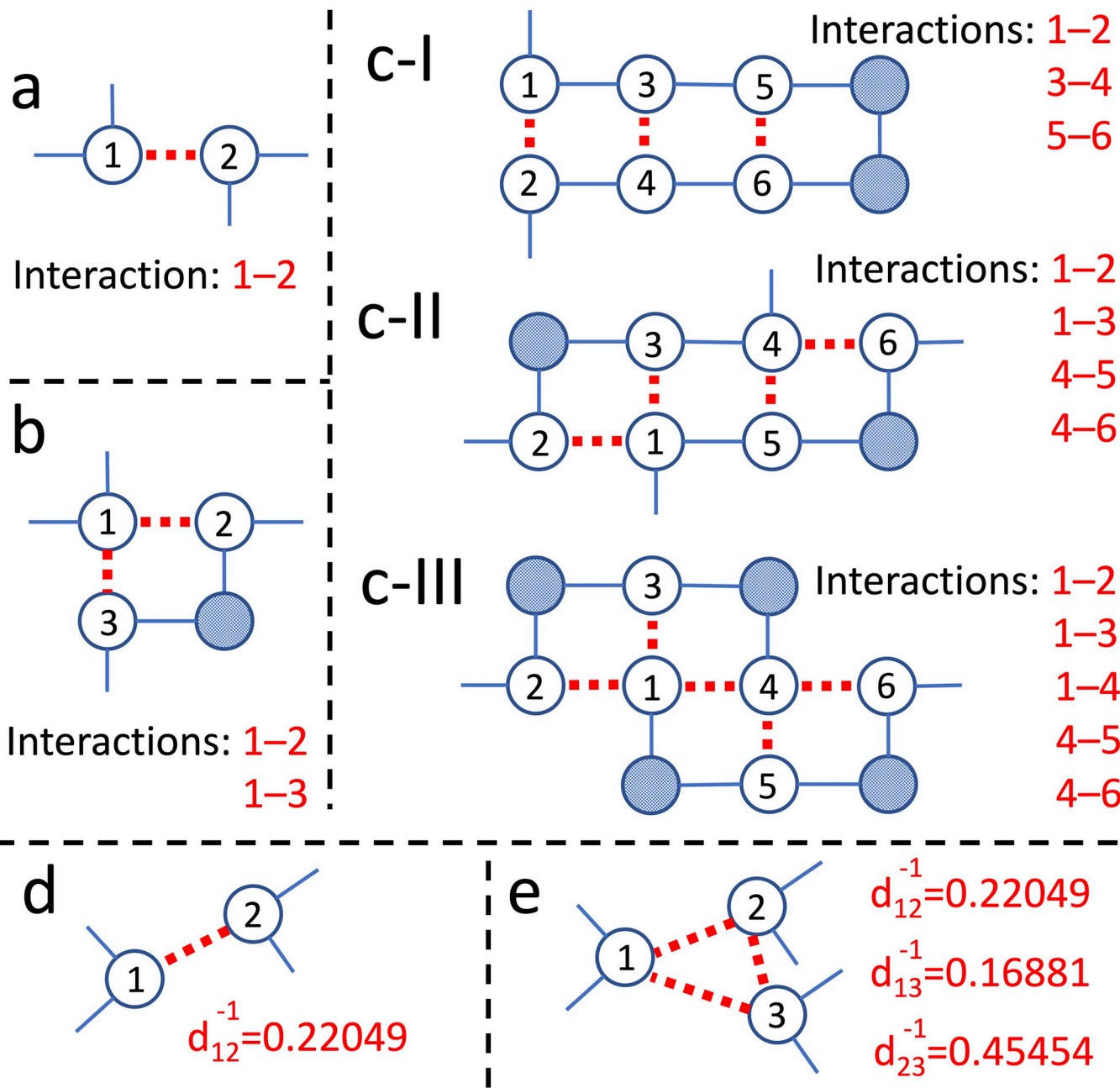

**Fig 2. Schematic representations of protein models.** SP model with: a) Two designable sites; b) Three designable sites; c) Six designable sites with I) three, II) four, and III) five pair-wise interactions. MR model with: d) Two designable sites; e) Three designable sites. The designable sites are shown as circles with numbers. The red dashed lines represent the interactions among the designable sites. In a–c) the pattern of interactions between the sites are provided for each structure. The checkered circles are non-interacting residues. *In this study, there are no geometrical differences between residues, and all are being treated as identical beads represented with circles.* In d) and e) the $d_{ij}^{-1}$ is the corresponding distance reciprocal between designable sites $i$ and $j$.

of the SP model, addressed as "IBM-SP". This model uses the hydrophobic-polar energy table in Fig 3D and the protein structure in Fig 2A.

Note that in the MR model, the $d^{-1}$ mainly acts as weights of interactions, providing a weighted impact of each pair-wise interaction in the system. However, considering $E_{coulomb} = \frac{kq_iq_j}{d}$, where $k$ is the Coulomb constant, $q$ is the electric charges, and $i$ and $j$ are the two

**a**

|    | H1 | H2 | Pol1 | Pol2 | Pos | Neg | X1 | X2 |
|----|----|----|------|------|-----|-----|----|----|
| **H1**   | -3 | -1 | +1 | 0 | +2 | +2 | 0 | 0 |
| **H2**   | -1 | -2 | 0 | +1 | +1 | +1 | 0 | 0 |
| **Pol1** | +1 | 0 | -3 | -2 | -1 | -1 | 0 | 0 |
| **Pol2** | 0 | +1 | -2 | -3 | +1 | 0 | 0 | 0 |
| **Pos**  | +2 | +1 | -1 | +1 | +4 | -4 | 0 | 0 |
| **Neg**  | +2 | +1 | -1 | 0 | -4 | +3 | 0 | 0 |
| **X1**   | 0 | 0 | 0 | 0 | 0 | 0 | -1 | 0 |
| **X2**   | 0 | 0 | 0 | 0 | 0 | 0 | 0 | 0 |

**b**

|    | H1 | H2 | Pol1 | Pol2 | Pos | Neg | X1 | X2 |
|----|----|----|------|------|-----|-----|----|----|
| **H1**   | -3.2 | -1.3 | +1.1 | +0.59 | +2.45 | +2.5 | 0 | 0 |
| **H2**   | -1.3 | -2.2 | 0 | +1 | +1.325 | +1.5 | 0 | 0 |
| **Pol1** | +1.1 | 0 | -3.7 | -2 | -1 | -1 | 0 | 0 |
| **Pol2** | +0.59 | +1 | -2 | -3.51 | +1.5 | 0 | 0 | 0 |
| **Pos**  | +2.45 | +1.325 | -1 | +1.5 | +4.2 | -4.3 | 0.5 | 0 |
| **Neg**  | +2.5 | +1.5 | -1 | 0 | -4.3 | +3.3 | 0 | 0 |
| **X1**   | 0 | 0 | 0 | 0 | 0.5 | 0 | -1.5 | 0 |
| **X2**   | 0 | 0 | 0 | 0 | 0 | 0 | 0 | 0.1 |

**c**

| Res. | Binary ID | Res. | Binary ID |
|------|-----------|------|-----------|
| H1   | → 000 | Pos | → 100 |
| H2   | → 001 | Neg | → 101 |
| Pol1 | → 010 | X1  | → 110 |
| Pol2 | → 011 | X2  | → 111 |

**d**

|    | H | P |
|----|----|----|
| **H** | -1 | 0 |
| **P** | 0 | 0 |

| Res. | Binary ID |
|------|-----------|
| H | → 0 |
| P | → 1 |

**Fig 3. Energy tables to represent the pair-wise interactions in our systems and the binary representation of residues.** Energy tables for: a) The SP model; b) The MR model. c) Binary representation of residues in the energy table. d) Energy table and the residue representations for the HP protein model. In a–c) *H1* and *H2* represent two types of hydrophobic residues, *Pol1* and *Pol2* represent two types of polar residues, *Pos* represents a positive residue, *Neg* represents a negative residue, and two types of "other" residues that do not fit in any of the previous categories are represented by *X1* and *X2*. Note that all energies in our tables have qualitative values. In d) *H* represents hydrophobic and *P* represents polar residues.

interacting particles, one can treat the values in Fig 3B as constants representing $kq_iq_j$ between residues $i$ and $j$. Thus, the MR model is directly implementing the Coulomb potential for protein design in its current form. Moreover, since we feed the pre-computed values/tables to our algorithms, the MR model can implement all potential energies for protein design in conventional packages, such as Rosetta biomolecular modelling suite [26]. For example, one can input the cosine value of distances or angles for the bonded or angular interactions.

Our results show that by using the quantum simulators, our circuits offer the expected $\mathcal{O}(\sqrt{N/M})$ queries to find the answers states, which confirm the utilization of Grover's algorithm's properties in them. However, the results of real quantum computers indicate the need for devices with much lower noise to implement our circuits.

## 2. Results

### 2.1. Number of qubits in the circuit

The number of qubits required to represent a residue in a unique binary format is given by $g = \lceil log_2(A) \rceil$. Thus, g = 3 for the eight amino acids used in both SP and MR models (Fig 3A, 3B and 3C). Even though our energy tables with eight types of residues are simpler than the canonical model with A = 20 (g = 5), they are still more complicated than the widely used HP table in the IBM-SP model (g = 1).

Table 1 shows the number of qubits required by different circuits in our study. The total number of n qubits in the circuits is given by

$$n = g \times s \tag{1}$$

where $g = log_2(number\ of\ residues)$. For the SP and MR models, $n = 3 \times s$, while this number for the IBM-SP model is $n = 1 \times s$.

In addition, we require m number of qubits in the circuit set to represent each numeric value, e.g., the energy of each pair-wise interaction, as a part of work qubits (Fig 1). The required number of qubits to represent a numeric value can be found as

$$m = \lceil log_2((|E_{max}| + |E_{min}|) \times i \times 2) \rceil + p \tag{2}$$

where $i$ is the number of interactions in the system, $p$ is the number of qubits allocated to represent the values after the decimal point, and $E_{min}$ and $E_{max}$ are the minimum and maximum values in the energy table in Fig 3. In the SP model, $p = 0$ and $i$ is represented as input to the

**Table 1. A brief description of each circuit in this study.**

| Circuit (model) | n | N | m | Oracle's work qubits | Total qubits (q) |
|---|---|---|---|---|---|
| s = 2 (SP) | 6 | 64 | 4 | 9 | 15 |
| s = 3 (SP) | 9 | 512 | 5 | 11 | 20 |
| s = 6,i = 3 (SP) | 18 | 262,144 | 6 | 13 | 31 |
| s = 6,i = 4 (SP) | 18 | 262,144 | 6 | 13 | 31 |
| s = 6,i = 5 (SP) | 18 | 262,144 | 7 | 15 | 33 |
| s = 2 (MR) | 6 | 64 | 9 | 145 | 151 |
| s = 2 (MR-MP) | 6 | 64 | 14 | 225 | 231 |
| s = 3 (MR) | 9 | 512 | 9 | 145 | 154 |
| s = 3 (MR-MP) | 9 | 512 | 14 | 225 | 234 |
| s = 2 (IBM-SP) | 2 | 4 | 2 | 5 | 7 |

circuit (Fig 2A). However, in the MR model

$$i = i_{max} = \left( \frac{s \times (s-1)}{2} \right) \tag{3}$$

Here, since we use fixed-point decimal numbers, setting $p = 5$ in Eq 2 provides the precision of 0.03125 that is the default for MR model. We also study this model with more precision (MR-MP), where $p = 10$ and the circuit can represent values smaller than ~0.001. More detail on how to choose the minimum required number of $m$ qubits for each system is provided in S1 Appendix.

As shown in Table 1, the number of work qubits is $2m+1$ and $16m+1$ for the SP and MR models, respectively. Since we use different calculations in the oracle, the required number of work qubits differs for the two models (discussed in the Methods section).

From Eq 1, Eq 2 and Eq 3, we have that the maximum of total number of qubits ($q$) grows as $\sim(3 \times s + c_1 log_2(s) + c_2)$, where $c_1$ and $c_2$ are constant values. Trying to calculate the distance reciprocals on the same quantum circuit, as discussed by Bhaskar *et al.* [27], would add $\mathcal{O}(s^2 - c_3 \times s)$ qubits to the circuit, where $c_3$ is a positive constant value.

Moreover, since the number of qubits required for the SP and MR model is large, we use the matrix product state (MPS) simulator [28], the only simulator that currently can simulate circuits with this many qubits. More detail is provided in the Methods section.

## 2.2. Finding the answer states

Fig 4 shows the probability of finding every individual state for four different systems containing two designable sites in the SP and MR models. Here, the answer states are clearly distinguished with their higher probabilities over the other states in the system. The answer states for all systems in this study are provided in detail in S2 Appendix.

In the SP model with two designable sites, by setting the $E_{th}$ to −3 (using Eq 6 in the Methods section), the algorithm finds the lowest number of answer states. Here, the algorithm finds two distinct results: *Pos-Neg* (i.e., residue 1 is *Pos* and residue 2 is *Neg*) and *Neg-Pos*, as shown in Fig 4A and 4B. These two results are expected as they are the lowest values in the pair-wise energy table with $E = -4$ in Fig 3A. Increasing $E_{th}$ to −2, the circuit finds five answer states, adding three new states to the previous two answer states from the $E_{th} = -3$ (Fig 4C and 4D).

Unlike the SP model, in the MR model, since we use decimal numbers, we can choose the $E_{th}$ value more precisely (Eq 7 in the Methods section). To choose the answer states within the *5%* and *15%* range of the minimum energy of the system, the $E_{th}$ is set to $95\%E_{min}$ and $85\%$ $E_{min}$, respectively. Here, the $E_{th} = 95\%E_{min}$ leads to two answer states, while choosing the $E_{th} = 85\%E_{min}$ provides three answer states (Fig 4E–4H).

The results in Fig 4 show that even with a single iteration ($R = 1$), finding each answer state is more probable than finding each non-answer state for $s = 2$ systems. However, the total probability of finding the answer states is less than the total probability of finding the non-answer states at $R = 1$, which is generally correct unless for systems with the maximum number of iterations ($R_{max}$) value of 1 or 2. In other words, to have a higher probability of picking the answer states (no matter which one) among all N possible states, the total probability of finding the answer states should be higher than *50%*. This probability increases to ~*100%* by increasing the number of iterations to $R = R_{max}$ (Eq 8 in the Methods section). For example, the total probability of answer states in the $s = 2, E_{th} = -3$ system in the SP model is only ~*26%* (~*12.9%* each state), while the probability of finding non-answer states is ~*74%* (Fig 4A). Nevertheless,

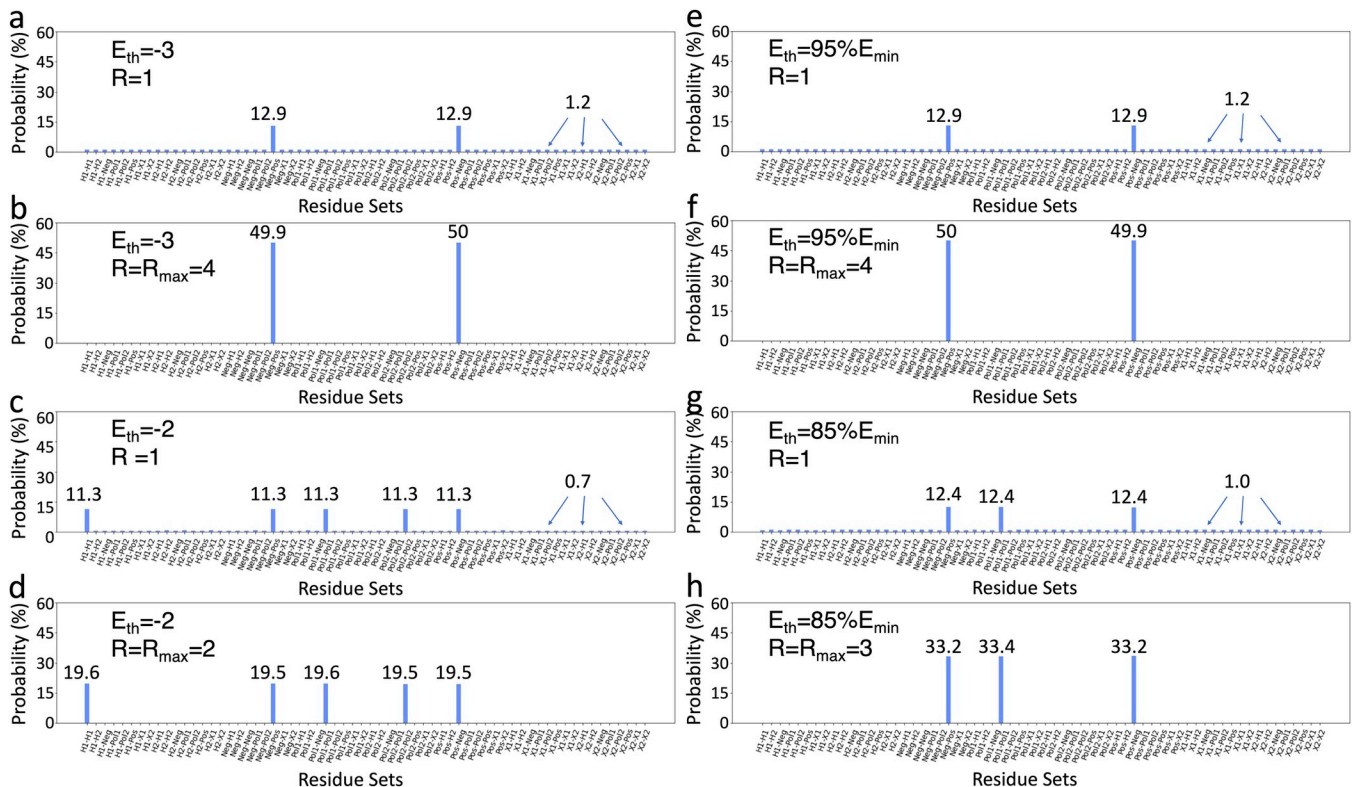

**Fig 4. Histogram representations of the probability of finding each state (64 in total) in circuits with two designable sites.** Results for circuits in the SP model with: a) $E_{th}$ = -3 and R = 1; b) $E_{th}$ = -3 and R = $R_{max}$ = 4; c) $E_{th}$ = -2 and R = 1; d) $E_{th}$ = -2 and R = $R_{max}$ = 2. Results for circuits in the MR model with: e) $E_{th}$ = 95%Emin and R = 1; f) $E_{th}$ = 95%$E_{min}$ and R = $R_{max}$ = 4; g) $E_{th}$ = 85%$E_{min}$ and R = 1; h) $E_{th}$ = 85%$E_{min}$ and R = $R_{max}$ = 3.

using $R = R_{max} = 4$, the probability of finding the answer states increases to ~100%, and the probability of all other states becomes almost zero (Fig 4B).

It should be noted that the results for the $s = 2, E_{th} = -3$ system in the SP model (Fig 4A and 4B) are almost identical to the results of the $s = 2, E_{th} = 95\%E_{min}$ system in the MR model (Fig 4E and 4F). Regardless of the oracle complexity in each model, since these two systems have the same number of answer states (M = 2) out of the same number of total states (N = 64), the probability of finding each state is the same for both systems. Moreover, since the *M* and the *N* are the same for these systems, they have the same behaviour with changing the number of iterations, which will be discussed in more detail later in this section.

In the case of systems with $s = 2$, the only recognizable difference between the SP and the MR models is the higher accuracy in choosing the $E_{th}$. Here, the role of the $d^{-1}$ in the MR model is suppressed due to the spatial symmetry in the two designable site systems. However, the $s = 3$ system with different distances between each site (Fig 2E) breaks the symmetry and illustrates the effect of $d^{-1}$ in the MR model. To show this effect, we compare the results for the structure shown in Fig 2E with a similar system with complete spatial symmetry, i.e., an equilateral triangle, where the $d_{ij} = 1$. The same answer states are provided for the $s = 3, E_{th} = 70\%$ $E_{min}$ *system* with the symmetrical and asymmetrical configurations. The same is correct for the $s = 3, E_{th} = 80\%E_{min}$ *system*. Nevertheless, by choosing the $E_{th} = 50\%E_{min}$, the system with the symmetrical configuration has 10 answer states while the asymmetrical system produces 19 answer states (results provided in S2 Appendix).

## 2.3. Role of number of iterations

Fig 5 shows the probability of finding the answer states as a function of the normalized number of iterations, $R/R_{max}$. Here, the probability curves represent a universal pattern for different systems in the SP and the MR models, whereby increasing the $R$, the probability of finding the answer states increases, reaching ∼ *100%* at $R = R_{max}$. Since the computational cost of simulating larger systems is high, only the first few iterations are simulated for *s = 6* systems in the SP model (Fig 5A), while running them for $R = R_{max}$ would require years of simulation on CPU and terabytes of RAM (discussed in S3 Appendix in detail). Moreover, the probability curves in Fig 5 follow the ~$sin^2(\alpha R)$, which is expected behaviour of Grover's algorithm [3,29], where $\alpha$ is a constant value (more detail in S4 Appendix).

The $R_{max}$ values obtained from simulations of systems in both SP and MR models, i.e., the ones reached to the $R/R_{max} = 1$ (Fig 5), are plotted against $N/M$ in Fig 6. These results show that the circuits follow the $R_{max} \propto \sqrt{N/M}$ behaviour, the quantum advantage that Grover's algorithm is expected to provide.

Moreover, the results for the $SP, s = 2, E_{th} = -3$ system and the $MR, s = 2, E_{th} = 95\%E_{min}$ system show that since $N/M$ values are the same, the $R_{max}$ values are identical (Fig 6). The same is correct for the $SP, s = 2, E_{th} = -2$ and the $MR, s = 2, E_{th} = 70\%E_{min}$ systems.

## 2.4. Number of gates used for classic and quantum algorithms

To compare conventional classical methods with our quantum circuits, assume we have a classical algorithm designed to search through all possible states to find the answers. Furthermore, suppose the same input data we use in quantum circuits are provided for the classic models, i.e., protein structures, interaction patterns, pre-computed pair-wise energy tables and distance reciprocals. Also, assume that the same number of bits and qubits are allocated to represent a value in both classic and quantum algorithms, i.e., $m$ is the same. The latter confirms that the number of computations and the accuracy of the calculated values are similar. Despite longer bits/qubits providing higher accuracy, they require more computations (gates) units. Note that even though the total number of qubits in our quantum circuits is limited to $q$ (Table 1), the total number of bits in the classical approach is not limited.

As discussed earlier in this paper, a classical search algorithm requires $\mathcal{O}(N)$ iterations to find the answer states. In contrast, our quantum circuits only require $\mathcal{O}(\sqrt{N})$ iterations. Nevertheless, this comparison does not consider the number of computations used in the classic and quantum approaches. In our quantum algorithms, the total number of computations is given by

$$\#ofQ_{tot} = \#ofQ_{init.} + \mathcal{O}(\sqrt{N}) \times (\#ofQ_{orcl.} + \#ofQ_{diff.}) \qquad (4)$$

where $\#ofQ_{init.}$, $\#ofQ_{orcl.}$, and $\#ofQ_{diff.}$ are the number of computations conducted in the initialization step, the oracle and the diffuser, respectively (Fig 1A). However, in the classic realm, the total number of computations ($\#ofC_{tot}$) required to go through all possible states to find the answers is

$$\#ofC_{tot} = \mathcal{O}(N) \times (\#ofC) \qquad (5)$$

where $\#ofC$ represents all computations required to find the energy of a single state.

For simplicity, we compare the SP model with a similarly complex classic model, referred to as "SP-classic". In the SP-classic model, similar to the SP model, the pair-wise energies of designable sites are added together for each $N$ combination of amino acids (sequence) and subtracted from the $E_{th}$ to find the answer states. In the classic computation, once the addition

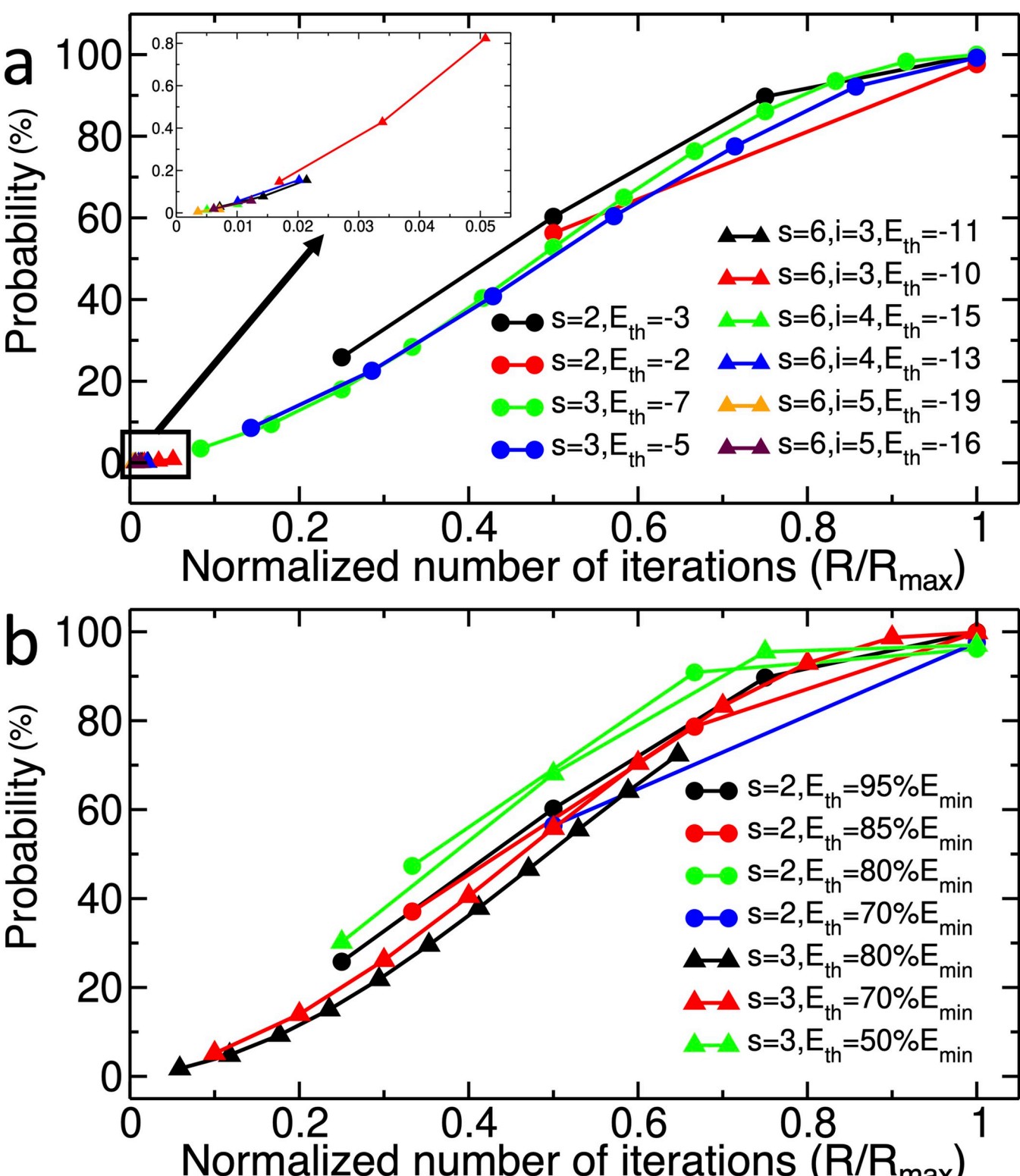

**Fig 5. Probability of finding answer states in different systems as a function of normalized number of iterations ($R/R_{max}$).** Results for a) The SP model; b) The MR model; The inset in a), represents data for the first few $R$ for systems with six designable sites.

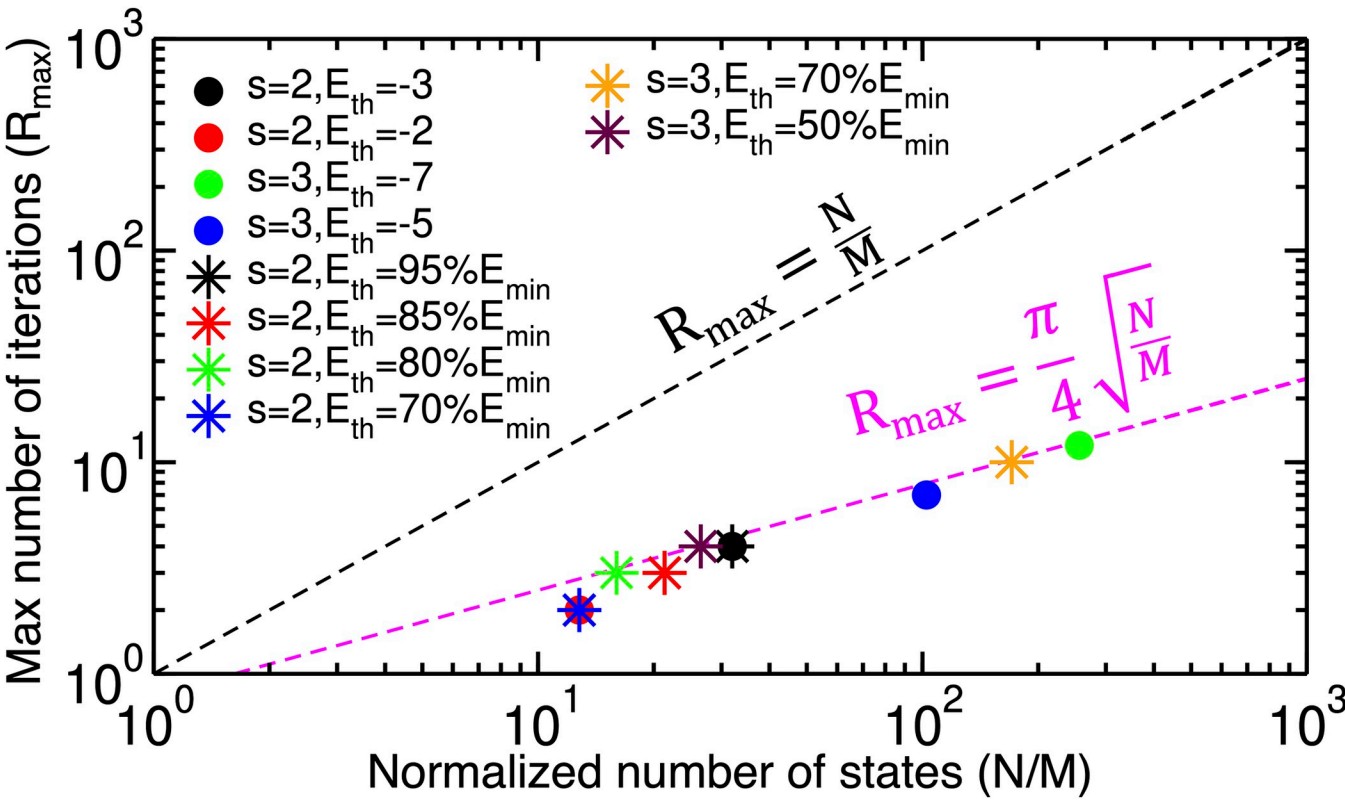

**Fig 6. The maximum number of iterations ($R_{max}$) values as a function of normalized number of states ($N/M$).** The circles represent the data for the SP model and the stars are the data for the MR model. The magenta dashed line shows the $R_{max}$ threshold for Grover's algorithm, while the black dashed line is the threshold of the classic realm.

between two numbers occurs in the SP-classic, the bits get restored and ready for the next number. However, for the SP model, due to the quantum nature of the circuit, the qubits that will be re-used should be "cleaned" by re-applying the gates (i.e., the computation is doubled compared to the SP-classic case). Moreover, the oracle in the quantum circuit should be cleaned, meaning all the gates should be re-applied (Fig 1B).

Details of calculating the number of computations for each step of circuits in the SP and SP-classic models are provided in S5 Appendix. For the SP model, the number of computations in the initialization step is $ofQ_{init.} \sim \mathcal{O}(\log_2(N))$ gates. The oracle cost ($\#ofQ_{orcl.}$) can be broken into the cost of introducing the energies, cost of adders, cost of subtracting the $E_{tot}$ from the $E_{th}$, and the negation cost. For introducing the energies, the total cost changes as $\sim \mathcal{O}[(\log_2(\log_2(N)) \times (\log_2(N))^2)]$. Moreover, it is shown that the number of gates required in adder functions is $\mathcal{O}(m)$ [30,31], requiring $\sim \mathcal{O}[\log_2(\log_2(N)) \times ((\log_2(N))^2]$ gates to compute the $E_{tot}$. Furthermore, the cost of subtracting the $E_{tot}$ from the $E_{th}$ is $\sim \mathcal{O}[\log_2(\log_2(N))]$, and the negation cost is $\sim \mathcal{O}(1)$. Thus, $ofQ_{orcl.} \sim \mathcal{O}[\log_2(\log_2(N)) \times \{(\log_2(N))^2 + \log_2(N) + 1\}]$. Finally, in the diffuser, the number of computations is proportional to $\sim \mathcal{O}(\log_2(N))$. Thus, using Eq 4 to find the total computation cost of computation, we have $ofQ_{tot} \sim \mathcal{O}[(\log_2(N)) + \sqrt{N} \times \{[\log_2(\log_2(N)) \times ((\log_2(N))^2 + \log_2(N) + 1)] + \log_2(N)\}]$.

For the SP-classic model, we ignore the cost of introducing the energies to the classic circuits due to the lack of information on this part. Thus, we only consider the cost of adding the

energy values to find the $E_{tot}$ and subtracting it from $E_{th}$ to find the answer states. In this case, the cost of the computation for the SP-classic model from Eq 5 is

$ofC_{tot} \sim \mathcal{O}(N \times [\log_2(\log_2(N)) \times (\log_2(N) + 1)])$.

These estimations show that for $N>56$, the number of computations in classic circuits is larger than the quantum algorithm, despite ignoring the cost of introducing the energy values to the classic circuit. Note that for the smallest case of $s = 2$, $N$ is 64, indicating that the number of computations in classic form is higher than the quantum circuit for all systems. Similar is correct for the MR model, which is discussed in more detail in S5 Appendix.

## 2.5. Real devices and the effect of noise in simulators

The IBM-SP model circuit of our algorithms implemented on real quantum devices has four possible answer states, i.e., *HH*, *HP*, *PH* and *PP*. As expected from the energy table (Fig 3D), by setting the $E_{th} = 0$ and using the ideal noise-free QASM [32] simulator, the circuit finds the HH state as the answer (Fig 7A).

In addition to the ideal simulations, we implement selective noise properties of the *ibmq_toronto* (Fig 7B-I) and the *ibmq_montreal* (Fig 7C-I) devices in the QASM simulator. Detail is provided in the Methods section. Implementing the gate, measurement and initialization fidelities, as well as the qubit connection mapping properties of the *ibmq_toronto* and the *ibmq_montreal* devices show that the simulations predict the expected answer state with the average probabilities of ~53.7% and ~33.5%, respectively. These results show that by only considering the selected noise sources, the real quantum devices are expected to distinguish the answer state from the others. Moreover, since the *ibmq_toronto* and the *ibmq_montreal* have the same qubit connection mappings, the large difference between the predicted results in Fig 7B-I and c-I implies lower gate, measurement and initialization fidelities for the *ibmq_toronto* compared to the *ibmq_montreal* quantum computer. These results suggest that despite having lower quantum volume (QV) [33], the *ibmq_toronto* device is more likely to distinguish the answer state clearly.

As expected from the results of the noise-included simulations, running the circuit on the *ibmq_toronto* device provides the answer state with the highest average probability of ~28.8%, while the *ibmq_montreal* device provides almost the same probabilities for all four states (Fig 7B-II and 7C-II). The probability of finding the *HH* state using the *ibmq_toronto* device is 28.77±2.39%, while for the second probable state, i.e., *HP*, this value is 24.94±1.26. These results indicate that the *HH* state has a higher probability considering the standard deviations. Note that despite having a higher probability, the probability of finding the answer states is only ~28.8%, while the probability of finding non-answer states is ~71.2%.

The difference between the results of the simulations considering the gates fidelities and the qubit mappings (Fig 7B-I and 7C-I) and the real quantum devices (Fig 7B-II and 7C-II) show the role of the performance parameters, such as coherence, crosstalk and spectator errors on the probability of the states. In the case of the *ibmq_toronto*, these performance parameters cause a significant loss in the probability of finding the answer state. These results are in agreement with the expected results of quantum devices in the NISQ era. As Chia *et al.* [34] and Li *et al.* [35] have discussed, circuits that could run on currently available real quantum devices are mostly limited to the circuit depth of only a few tens of gates, with a maximum of ~55. These studies suggest that the depth of our circuit transpiled on real quantum devices (i.e., ~160) is much larger than the current limit of the NISQ era quantum computers. Thus, even though the noise-included simulators predict the answer states with a high probability (Fig 7B-I and 7C-I), the "large" circuit depth hinders the answer state on real quantum devices (Fig 7B-II and 7C-II).

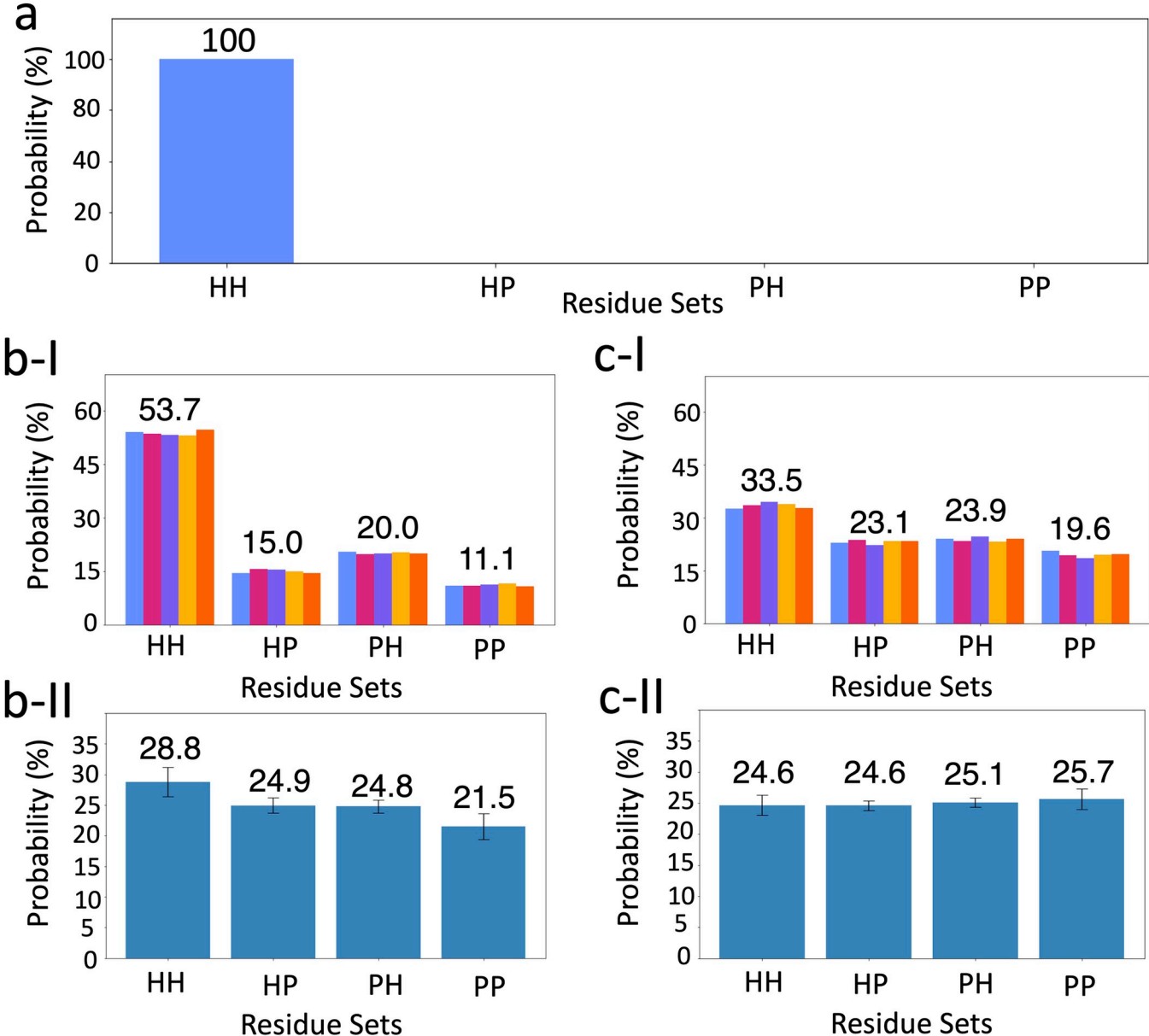

**Fig 7. Histogram representations of probabilities for each state in the IBM-SP model circuit.** a) Results of the circuit, using the ideal MPS quantum computer simulator. b) Results of the circuit using: I) gate fidelity, measurement fidelity, initialization fidelity and qubit mapping of *ibmq_toronto* in the simulator; II) Real *ibm_toronto* device averaged over 20 different runs. c) Results of the circuit using: I) gate fidelity, measurement fidelity, initialization fidelity and qubit mapping of *ibmq_montreal* in the simulator; II) Real *ibm_montreal* device averaged over 30 different runs. In b) and c), the data bars with different colours show results for five separate runs. The error bars represent the standard deviation from the mean value. The numbers on each set of bars show the average probability of the state. The number of samplings for all plots is set to 8,192 shots.

## 3. Discussions

This work studies developing gate-based circuits to address protein design problems by implementing a pure quantum computing algorithm, i.e., Grover's algorithm. Using ideal quantum computer simulators shows that our quantum circuits can find the $M$ desired answer states among $N$ total states for systems with different complexities. Furthermore, the results confirm that using the maximum number of iterations ($\mathcal{O}(\sqrt{N/M})$) provides the maximum

probability of finding the answer states (~100%), indicating the quadratic advantage of our approach over classical methods on conventional computers in search for answer states.

Moreover, the maximum number of computations required in our quantum circuits is smaller than the least number of computations required for classical models, confirming the quantum advantage of our circuits over conventional methods. Furthermore, our results show that the MPS simulator can correctly simulate highly entangled algorithms such as Grover's algorithm, with a large number of qubits in the circuit (up to 234 qubits).

In this work, the simplified model, i.e., the SP model, implements protein configurations similar to the 2D square lattice model, using integer numbers for calculating the energies. However, due to the limitation of the CPU times and computational resources, the largest circuits are limited to systems with six designable sites, simulating a simple hair-pin protein structure (Fig 2C-I) or complex and compact intra-protein configurations (Fig 2C-II and 2C-III). Adding more complexity to the system in the MR model, i.e., introducing the distances reciprocals and decimal numbers, limits the number of simulatable designable sites in a system to three. However, this model enables us to use more realistic energy terms in the Hamiltonian of a protein design problem on quantum computers, i.e., mimicking the Coulomb energies. Nonetheless, all different types of potentials used in conventional protein design methods could similarly be implemented in our algorithm. Moreover, our approach in implementing the pre-calculated distances reciprocals and decimal numbers and using multiplication functions could also be employed in the protein folding studies with quantum computers currently limited to uniform lattice models [11–14].

At the current NISQ stage, the number of available qubits, their connectivity and the noise associated with using each gate limit studying pure quantum algorithms on real quantum devices. In these devices, since the depth of circuits is restricted to a few tens of gates, even running a small protein design circuit (i.e., IBM-SP) does not provide a definitive result due to the noise and requires further improvements in quantum computers. Moreover, for the SP,s = 2 system, the depth of the circuit in a fully connected simulator is ~400 gates, while after transpilation, the depth of the circuit on real quantum devices could reach ~17,000 gates. As an alternative approach, it would be of interest to use popular hybrid approaches in the NISQ era, e.g., the QAOA method, to study the protein design problem and compare results with those provided by circuits using Grover's algorithm in the simulations. Nevertheless, improving the quantum devices at the current pace encourages us that in the near future, our circuits can be implemented on real quantum computers and show the advantage of quantum computers in protein design problems.

## 4. Methods

### System setup & simulations

Following the Grover's algorithm in Fig 1, after the initialization, the oracle is programmed to implement energies and conduct required calculations to find and mark the answer states. The oracle has different sub-steps associated with it. First, the classically-calculated values in the energy tables (Fig 3) are introduced to the oracle, based on the protein structure and the interaction pattern (Fig 2). Implementing the pair-wise energies in the oracle is described in the S6 Appendix. Next, the energy values for every pair-wise interaction in the structure are summed using quantum computing adders to find the total energy, $E_{tot}$, for each sequence of residues (i.e., each state). This part of the oracle is the only part of the algorithm with different setups for the SP and the MR models, which is discussed in detail later in this section. Then, the $E_{tot}$ is subtracted from a threshold energy value of the circuit for each state, and the ones with the negative result are the answer. The $E_{th}$ is explicitly set for each circuit, and by changing its

value, the answer states change. Details on how the $E_{tot}$ and $E_{th}$ are calculated for the SP and MR models will be discussed later in this section. Finally, the oracle negates the amplitude of the answer states. After marking the answer states (i.e., the negation), the oracle un-computes all previous steps (except for the negation step) to clear the work qubits and prevent them from affecting the final results [36].

In the SP model, the oracle calculates the total energy of the state $k$ (out of $N$ states) using:

$$E_{tot-SP}(k) = \sum_{a>b} E_{a,b}(k),$$

where $E_{a,b}(k)$ is the energy value of the interaction between designable sites $a$ and $b$ in the structure (Fig 2A–2C), while specific residues in the set $k$ fill these sites (Fig 3A). Here, since there are no distance dependencies, all pair-wise interactions contribute equally to the total energy of the system. Moreover, the lowest $E_{th}$ value in the SP model is defined as:

$$E_{th-SP}(k) = (E_{min} \times i) + 1 \tag{6}$$

This $E_{th-SP}$ provides the lowest number of answer states for each circuit in the SP model (and similarly in the IBM-SP model).

In the MR model, the total energy of the state k in the oracle is calculated using:

$$E_{tot-MR}(k) = \sum_{a>b} E_{a,b}(k) \times d_{a,b}^{-1},$$

where $d_{a,b}^{-1}$ is a dimensionless matrix representing the distance reciprocal between designable sites $a$ and $b$. Thus, nearer designable sites have more contributions to the $E_{tot-MR}$. In the MR model, the $E_{th}$ is defined as:

$$E_{th-MR}(k) = B \times (E_{min} \times \sum_{a>b} d_{a,b}^{-1}) \tag{7}$$

where $B$ is a unitless constant decimal number, less than 1. For simplicity, if $B$ is set to 0.95 the $E_{th-MR}$ is referred to as $E_{th-MR} = 95\% E_{min}$ in this paper. Note that in this work, the $E_{th-SP}$ and $E_{th-MR}$ can be distinguished by their values, i.e., being equal to an integer number and being a percentage of the $E_{min}$, respectively. Thus, in the paper, we refer to both as $E_{th}$, removing the "SP" and "MR" subscripts.

Moreover, it should be noted that the $E_{th}$ value calculated for the SP model system (Eq 6) captures the lowest energy structures for different numbers of designable sites (see results in S2 Appendix). However, the $E_{th}$ value calculated for the MR model (Eq 7) does show the limit below which there are no answer states (for B = 1). Due to the presence of distance reciprocals and the fact that all designable sites interact in the MR model circuits, one should carefully set the B value to attain the desired results. As the number of designable sites increases, the B value and, equivalently, the $E_{th}$ value to find the state with the lowest energy decreases (see results in S2 Appendix for the MR,s = 2 and MR,s = 3 systems). Unlike the SP model, the $E_{th}$ in the MR model does not define the lowest energy states but a heuristic threshold from an empirical experience to find low energy states.

In the oracle, a version of the quantum ripple-carry adder introduced by Cuccaro et al. [30] is used for adding (and subtracting) the values for both SP and MR models, which requires $2m+1$ qubits to add numbers, each represented with $m$ qubits. This adder is also implemented as a part of the multiplication function employed in the MR model, using $16m+1$ qubits to multiply the two numbers.

The role of Grover's diffuser (Fig 1) is to act on the $n$ qubits and increase the probability of answer states over all the other states in the circuit. To accomplish this, the diffuser changes the negative amplitude of the answer states (marked in the oracle) to positive and then increases the amplitude of these flipped states [3]. Note that since the total probability of all

states is one, increasing the amplitude of the answer states (and thus the probability of finding them) decreases the amplitude of the non-answer states.

The final step of the algorithm is the measurement (Fig 1), which is done on all n qubits to find the M answer states among all N possible states. Note that the work qubits are not measured and are discarded.

In Grover's algorithm, the upper bound of the number of iterations required to get the answer states with the highest probability is:

$$R_{max} \leq \left\lceil \frac{1}{2} \frac{\pi}{2 \times arcsin\left(\sqrt{\frac{M}{N}}\right)} \right\rceil + \mathcal{O}\left(\sqrt{\frac{M}{N}}\right),$$

while, in the $N \gg M$ limit, the equation changes to:

$$R_{max} \leq \left\lceil \frac{\pi}{4} \sqrt{\frac{N}{M}} \right\rceil \tag{8}$$

We use IBM's Qiskit package [37] to generate the circuits and simulate them in this study. Better known simulators such as the QASM require large amounts of RAM for the simulations, scaling as $16 \times 2^q$, requiring at least 128 GB of RAM for the largest SP model circuit with q = 33 (Table 1). However, for circuits using more than 150 qubits in the MR model, this number reaches $\sim 2 \times 10^{34}$ TB of RAM. Thus, due to the unprecedented size of our systems, we use the MPS simulators as the only possible choice to simulate all circuits. We use the computational resources provided by the Cedar cluster [38] to run our circuits on the quantum computer simulator.

## Real quantum devices and noise-containing simulators

The quantum circuits in this work are composed of several single-qubit and multi-qubit (including two or more qubits) gates. Generally, several single-qubit and multi-qubit gates with low complexities are predefined in quantum computer simulators (details vary by simulator packages). However, in the case of more complex gates such as the *CC–NOT* (Toffoli) gate, the simulators decompose them into the simpler predefined gates.

Unlike simulators, real quantum devices only support a handful of native gates. Therefore, all gates are decomposed into the native gates when simulating our circuits on real quantum computers. However, depending on the type of the device and its manufacturer, these native gates may vary.

In addition to supporting only a few native gates, using the real quantum devices has further limitations. Since we are currently in the NISQ era, the qubits and gates contain noise while transferring data. Currently, IBM quantum devices have error rates of $\mathcal{O}(10^{-2})$ and $\mathcal{O}(10^{-4})$ for the *C-NOT* and single-qubit gates, respectively [39]. Moreover, IBM quantum devices have limited connectivity between qubits [39]. Thus, swap gates are required to perform two-qubit gates between not directly connected qubits, increasing the number of gates and the quantum computational cost of the circuit.

We use the circuit depth metric to measure and compare the complexity of circuits in this study. The circuit depth represents the number of gates in the longest path in a circuit [40], i.e., from the initialization to the measurement in our systems (Fig 1). For the circuit of two designable sites (Fig 1A) at *R = 1* in the SP model, the simplest system studied on quantum computer simulators in our study, the circuit depth in the decomposed stage on an ideal device with fully connected qubits is ~17,000 (more discussion is provided in S7 Appendix). In the

current NISQ era, it is impossible to run a circuit with such a "large" depth on a real device due to the noise introduced to the results, owing to the gate fidelities and the system decoherence for deeper circuits [25]. Note that the depth of the circuit will increase significantly if the qubits are not fully connected on a quantum computer.

The IBM-SP model, executed on real quantum devices, requires seven qubits in total with $n = 2$ (Table 1). By setting the $E_{th}$ to 0, the circuit will have one answer state (Fig 3D) and the $R_{max} = 1$. We employ two IBM quantum computers, the *ibmq_toronto* (v1.6.1 and v1.6.2) with the IBM Quantum Falcon r4 processors and the *ibmq_montreal* (v1.10.11) with the IBM Quantum Falcon r4 processors, both having 27 qubits and the same pattern of connectivity. To run the IBM-SP model on these devices, we use optimization level 2 and a selective pattern of qubits to transpile the ideal circuit on them. Moreover, we apply the transpilation 500 times for each run and select the circuit with the lowest depth, ranging between 158 and 167, as an input for the real quantum devices. To compare these quantum computers, we use the quantum volume, a unitless number, quantifying the largest random circuit that a quantum computer can implement successfully [33]; thus, the more, the better.

## Supporting information

**S1 Appendix. Number of required qubits in the circuit and the general algorithm.**
(PDF)

**S2 Appendix. Explicit results of each system.**
(PDF)

**S3 Appendix. Computational costs of the simulations on conventional computers.**
(PDF)

**S4 Appendix. Role of number of iterations.**
(PDF)

**S5 Appendix. Computational costs in the circuits.**
(PDF)

**S6 Appendix. Implementing energies in the circuit.**
(PDF)

**S7 Appendix. Computational Cost of Simulations and Circuit Depth.**
(PDF)

## Acknowledgments

The authors would like to thank WestGrid (www.westgrid.ca) and Compute Canada (www.computecanada.ca) for providing computational resources for this project.

We acknowledge the use of IBM Quantum services for this work. The views expressed are those of the authors, and do not reflect the official policy or position of IBM or the IBM Quantum team.

We also would like to acknowledge CMC Microsystems for facilitating this research, specifically through their member access to the IBM Quantum Hub at Institut quantique.

## Author Contributions

**Conceptualization:** Mohammad Hassan Khatami, Nathan Wiebe, Philip M. Kim.

**Data curation:** Mohammad Hassan Khatami, Udson C. Mendes.

**Formal analysis:** Mohammad Hassan Khatami.

**Funding acquisition:** Philip M. Kim.

**Investigation:** Mohammad Hassan Khatami.

**Methodology:** Mohammad Hassan Khatami, Nathan Wiebe.

**Resources:** Udson C. Mendes.

**Supervision:** Philip M. Kim.

**Validation:** Mohammad Hassan Khatami, Philip M. Kim.

**Visualization:** Mohammad Hassan Khatami.

**Writing – original draft:** Mohammad Hassan Khatami.

**Writing – review & editing:** Mohammad Hassan Khatami, Udson C. Mendes, Nathan Wiebe, Philip M. Kim.

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
