## [Decision Letter · Decision Letter 0]

8 Mar 2023

Dear Dr. Kim and Co-authors,

Thank you very much for submitting your manuscript "Gate-based quantum computing for protein design" for consideration at PLOS Computational Biology. As with all papers reviewed by the journal, your manuscript was reviewed by members of the editorial board and by several independent reviewers. The reviewers appreciated the attention to an important topic. Based on the reviews, we are likely to accept this manuscript for publication, providing that you modify the manuscript according to the review recommendations.

Please revise the manuscript to address the comments of both referees and resubmit.

We request that you make your code for building the quantum circuits for protein design in this paper publicly available. Please also make publicly available any other code or information that will allow other researchers to reproduce your results.

Sincerely,

Roman Krems

Guest Editor

PLOS Computational Biology

Nir Ben-Tal

Section Editor

PLOS Computational Biology

Please revise the manuscript to address the comments of both referees and resubmit.

We request that you make your code for building the quantum circuits for protein design in this paper publicly available. Please also make publicly available any other code or information that will allow other researchers to reproduce your results.

Reviewer's Responses to Questions

**Comments to the Authors:**

Comment by Section Editor:

My apology for the very long time that it took us to handle your manuscript. The topic is outside my own expertise and also far from the rest of our editorial board. Thus, I had to look for a guest editor, which took long, presumably because I was outside my comfort zone. 

Reviewer #1: See PDF attachment. (The text below duplicates the PDF.)

I have read the submission by Khatami, Mendes, Wiebe, and Kim titled “Gate-based quantum computing for protein design”. In this manuscript, the authors examine the optimization problem of finding an amino acid sequence corresponding to the minimum value of an energy or scoring function, a problem that is solved during the protein design process. This problem scales poorly with the number of amino acid positions being designed in the protein (which the authors represent as s), or with the number of amino acid types considered at each position (which the authors represent as A), and this represents a serious challenge for design algorithms on classical computers. The authors show that this problem can be solved on gate-based quantum computers using Grover’s algorithm, a method for searching a large solution space using quadratically fewer samples than would be required classically. Using quantum simulators run on classical hardware, they show that they are able to produce solution states with high probability (quantum computing is probabilistic computing) for small problems, and that noise hinders but does not eliminate the functionality of their approach. They also show enrichment of the solution state for a very small problem on an actual, current-generation IBM gate-based quantum computer.

The authors have identified a sensible problem for which quantum computers could plausibly offer an advantage over classical computers one day. Their approach appears to be sound. The manuscript reads well, like one that has had quite a bit of thought go into revisions already. I am largely satisfied that this manuscript is suitable for publication. Below, I list one concern that I hope the authors can address or rebut, as well as a small number of minor points related to presentation which I think the authors will be able to address easily.

Major concern:

1. The PLoS author guidelines state: “PLOS expects researchers to share software and scripts needed for the work. If this cannot be made publicly available (e.g. due to licenses), the simulation method should be provided in sufficient detail so the results can, in principle, be reproduced using publicly available software.” I think that the authors have made a good faith effort to describe their methods in a good deal of detail. I am moderately confident that a person with sufficient domain expertise could reproduce what was done here, though the complexity of the approach does mean that it would be hard to say for certain whether any differences that that person saw between their results and the authors’ results was due to errors in reimplementing the method versus actual failure of the method to reproduce the result. I would be more comfortable if the authors were willing to make their Qiskit code for simulating these circuits (or, at least, for simulating a particular problem) publicly available. I won’t insist on this, since the description in the supplement is quite detailed, but it would make me more comfortable for the sake of full reproducibility.

Minor points:

1. On lines 70-72, the statement of the main characteristic of NP-complete problems could be a bit more precise and a bit clearer. The main characteristic of these problems is that the time or resources needed to find a solution scales poorly, not that they are intrinsically costly across all problem sizes. (There are NP-complete problems, like the travelling salesman problem, that are solved routinely for small problem instances, such as getting delivery vehicles to customers.) Similarly, on line 73, the proposed answer need not be scored easily, but in polynomial time. (There are polynomial-time tasks, such as the two-body molecular docking problem, that are extremely costly.) I mention this mainly because this is often a misconception that I have to dispel with students, that NP-completeness intrinsically means computational intractability or that polynomial-time scaling intrinsically means computational tractability for all problem sizes. It’s actually something that’s kind of interesting about the protein design problem: it’s an NP-complete problem with interesting real-world applications across a range of scales, ranging from extremely tractable (allowing classical and quantum solvers to be compared on these problems) to challenging (meaning that there may be problems for which there’s a slight quantum advantage) to utterly intractable (meaning that a good quantum solver would have problems that it was uniquely suited to do).

2. In the caption to Figure 1, there’s a sentence that seems to be incomplete. (As an aside, I like the detail in figure 1 – particularly panel B. Explanations like this in plan language make these algorithms much easier to understand.)

3. On lines 137-138, “is not advantageous” is a little bit confusing. Maybe it could be made clearer that the sign flip does not, at this point, alter the relative probabilities of states (something that will be altered by the subsequent diffuser step)?

4. Figure 2 is pretty clear. A suggestion that the authors might consider is adding a dashed line for each pairwise interaction, maybe in the same dark red colour as the listed interactions. (The lists currently refer to a feature that the reader can infer, but which is invisible. I wonder if it might be a little easier on the reader just to make it more visible.)

5. On lines 194 and 201, “number of residues” is a bit ambiguous: it could be the number of designable positions, s, or the number of amino acid possibilities at each position, A. I think the latter is meant. Perhaps using the nomenclature established earlier (A) or indicating that this is number of residue types might make this clearer.

6. A small singular/plural typo on line 228: “simulate circuits with this many qubits.”

7. Figures 5 and 6 show good results. The one minor criticism I have is of the x-axis label in figure 5 and the x- and y-axis labels in figure 6: descriptive labels such as “Max number of iterations (Rmax)” are easier to follow than the symbol alone (especially when the axis label is otherwise an expression of several symbols, such as R/Rmax. It’s not always intuitive to a reader why a particular expression is on an axis or what the expected relationship between an abstract expression on one axis and an abstract expression on another should be.)

8. It’s worth being aware that a circuit depth of 20 (line 415) sounds a little bit low as an upper limit. My experience has been that circuit depth limits of 40 or 50 seem a little more realistic, currently. This is very minor, though: either way, the authors’ point that only very short circuits are currently possible is a valid one.

9. On line 444, “Coulomb” should be capitalized.

10. I’d be a little bit careful with the claim on lines 446-448, that the distance reciprocal and decimal number approach used here could be used in lattice model folding studies. If the claim is that one could keep the lattice model, but use pairwise interaction energies represented as decimal numbers, possibly taken from classically precomputed lookup tables, this sounds reasonable. If the claim is that this provides a route to non-integer positions for the beads on the chain (i.e. a means of moving past lattice models to continuous-space models), this is harder to envisage, and would require means of dealing with very non-pairwise effects (such as the fact that the position of residue 3 depends on both residue 1 and residue 2). Maybe rephrasing this to make it clear that the former is meant would be a good idea.

11. In the first paragraph of the methods, it might be good to make it even more explicit what’s done classically and what’s done in the quantum circuit. For instance, “First, the classically-computed values in the energy tables are introduced to the oracle… Next the energy values for every pair-wise interaction in the structure are summed in the quantum circuit to find the total energy…”

12. The point made on lines 549-555, about the circuit depth needed for a real problem compared to what’s currently possible, is an important one, and one that many readers will be looking for in the conclusions. It’s likely worth including in the discussion in more detail, not just in the methods. I suspect that many readers will be looking for this as a bottom line to the story: they’ll be wondering, “at what point is the quantum hardware likely to be useful for real design problems”? (Note: I do NOT consider the fact that real problems will require larger, more robust hardware than is currently available to be a shortcoming of this work.)

Reviewer #2: Summary:

This paper proposes to use Grover’s algorithm to solve the protein design problem, which asks to decide the amino acid type on each designable site to achieve the lowest energy. In contrast to prior gate-based hybrid quantum-classical computation and quantum annealing, this work presents a gate-based pure quantum algorithm solution. The main task in this work is to construct the oracle in Grover’s algorithm. With the oracle, Grover’s algorithm can be applied to find the desired solution. It shows that the expected quadratic speed-up can be achieved compared to classical brute-force search. Experimental justification is performed on simulators and real quantum devices to show the correctness of the method.

Strengths:

- The protein design problem is important and its effective solutions will be widely useful.

- The first gate-based pure quantum algorithm for the protein design problem is proposed.

- The method is flexible in customizing for different energy tables and protein structure models.

- Related work is adequately cited.

Weaknesses:

- The design of the oracle is somewhat brute-forced, leading to extremely high costs.

- The oracle construction, which is the main technicality of the paper, is not well described.

- There is no experimental comparison with prior methods.

Specific comments:

Grover’s algorithm is a well-known quantum algorithm. Therefore, the main contribution of this work is designing the oracle to make the algorithm applicable to the considered problem of protein design. However, the article does not focus on explaining the oracle construction, but rather spends too much effort introducing Grover’s algorithm. Moreover, the oracle seems to directly use existing structures of adders and multipliers, trying to do numerical operations directly on quantum circuits. This direct adoption without optimization leads to extremely high costs, limiting the problem to small examples (e.g., 6 designable sites) even for simulation-based justification. It would be helpful if optimization can be performed to minimize the oracle circuit.

The value of E_th may seem a heuristic coming from an empirical experience. Although Eq. (6) and Eq. (7) may result in a good performance (providing few answer states) in the tested benchmarks, their applicability and generality may be questionable. For example, if there are more designable sites, maybe 95% E_min may be unattainable for any candidate states, leading to empty solution space. It would be good to discuss how to set E_th. Moreover, it seems that this method only finds solutions with lower energy rather than the lowest energy.

Although related work is introduced, a more in-depth comparison should be performed to reveal the strengths and weaknesses of different approaches.

**Have the authors made all data and (if applicable) computational code underlying the findings in their manuscript fully available?**

Reviewer #1: **No: **See my first point in the PDF of my review. The authors have done a pretty good job of describing their methods, particularly in the supplement. Reproducing these methods based on the description would be possible, but it would be easy to make errors in the re-implementation. This makes difficult comparison of one's own results with the re-implemented method to the original method described here. I'd be more comfortable if Qiskit code implementing these quantum circuits were made publicly available.

Reviewer #2: **No: **It would be nice if the authors can make the quantum circuits of protein design in this paper publically available.

PLOS authors have the option to publish the peer review history of their article (what does this mean?). If published, this will include your full peer review and any attached files.

Reviewer #1: **Yes: **Vikram K. Mulligan

Reviewer #2: No

Figure Files:

Data Requirements:

Reproducibility:

References:

---

## [Editor Report · Decision Letter 1]

17 Mar 2023

Dear Professor Kim,

We are pleased to inform you that your manuscript 'Gate-based quantum computing for protein design' has been provisionally accepted for publication in PLOS Computational Biology.

Best regards,

Roman Krems

Guest Editor

PLOS Computational Biology

Nir Ben-Tal

Session Editor

PLOS Computational Biology

---

## [Editor Report · Acceptance letter]

5 Apr 2023

PCOMPBIOL-D-22-01706R1 

Gate-based quantum computing for protein design

Dear Dr Kim,

I am pleased to inform you that your manuscript has been formally accepted for publication in PLOS Computational Biology. Your manuscript is now with our production department and you will be notified of the publication date in due course.

With kind regards,

Anita Estes
